# Modeling and Forecasting Monkeypox Cases Using Stochastic Models

**DOI:** 10.3390/jcm11216555

**Published:** 2022-11-04

**Authors:** Moiz Qureshi, Shahid Khan, Rashad A. R. Bantan, Muhammad Daniyal, Mohammed Elgarhy, Roy Rillera Marzo, Yulan Lin

**Affiliations:** 1Department of Statistics, Shaheed Benazir Bhutto University, Nawabshah 67450, Pakistan; 2Department of Mathematics, National University of Modern Languages, Islamabad 44000, Pakistan; 3Department of Marine Geology, Faculty of Marine Science, King AbdulAziz University, Jeddah 21551, Saudi Arabia; 4Department of Statistics, The Islamia University of Bahawalpur, Bahawalpur 63100, Pakistan; 5The Higher Institute of Commercial Sciences, Al Mahalla Al Kubra 31951, Egypt; 6Department of Community Medicine, International Medical School, Management and Science University, Shah Alam 40100, Selangor, Malaysia; 7Global Public Health, Jeffrey Cheah School of Medicine and Health Sciences, Monash University Malaysia, Jalan Lagoon Selatan, Subang Jaya 47500, Selangor, Malaysia; 8Department of Epidemiology and Health Statistics, School of Public Health, Fujian Medical University, Fuzhou 350122, China

**Keywords:** time series data analysis, monkeypox, ARIMA, MLP, pandemic

## Abstract

Background: Monkeypox virus is gaining attention due to its severity and spread among people. This study sheds light on the modeling and forecasting of new monkeypox cases. Knowledge about the future situation of the virus using a more accurate time series and stochastic models is required for future actions and plans to cope with the challenge. Methods: We conduct a side-by-side comparison of the machine learning approach with the traditional time series model. The multilayer perceptron model (MLP), a machine learning technique, and the Box–Jenkins methodology, also known as the ARIMA model, are used for classical modeling. Both methods are applied to the Monkeypox cumulative data set and compared using different model selection criteria such as root mean square error, mean square error, mean absolute error, and mean absolute percentage error. Results: With a root mean square error of 150.78, the monkeypox series follows the ARIMA (7,1,7) model among the other potential models. Comparatively, we use the multilayer perceptron (MLP) model, which employs the sigmoid activation function and has a different number of hidden neurons in a single hidden layer. The root mean square error of the MLP model, which uses a single input and ten hidden neurons, is 54.40, significantly lower than that of the ARIMA model. The actual confirmed cases versus estimated or fitted plots also demonstrate that the multilayer perceptron model has a better fit for the monkeypox data than the ARIMA model. Conclusions and Recommendation: When it comes to predicting monkeypox, the machine learning method outperforms the traditional time series. A better match can be achieved in future studies by applying the extreme learning machine model (ELM), support vector machine (SVM), and some other methods with various activation functions. It is thus concluded that the selected data provide a real picture of the virus. If the situations remain the same, governments and other stockholders should ensure the follow-up of Standard Operating Procedures (SOPs) among the masses, as the trends will continue rising in the upcoming 10 days. However, governments should take some serious interventions to cope with the virus. Limitation: In the ARIMA models selected for forecasting, we did not incorporate the effect of covariates such as the effect of net migration of monkeypox virus patients, government interventions, etc.

## 1. Introduction

After more than two years of serious economic and health crises, COVID-19 will soon likely enter an endemic stage. However, concerns about the occurrence of one viral after another have reached a fever pitch. The world is facing a second new viral outbreak-the monkeypox outbreak. The “monkeypox virus” (MPV) the causative agent of monkeypox is not new, as it was first discovered in 1958 in Copenhagen [1]. However, the first documented case of MPV was in a nine-month-old child from the Democratic Republic of Congo (DRC) in 1970 [2]. Since then, the outbreaks have risen but are primarily limited to the African continent. However, a limited spread to Europe and North America was also noted [3]. More than 48 confirmed cases in six different African countries from 1970 to 1979 were observed, including 38 cases in DRC, 4 in Liberia, 3 in Nigeria, and single cases in Cameroon, and Cote d’Ivoire. By 1986 the total cases reached 400 with mortality approaching 10%. Similarly, small outbreaks in equatorial Central and West Africa were also observed [4], including 500 cases in DRC alone between 1991 and 1999 [5]. Since the MVP has been in decline or reached an endemic situation in the African continent.

However, once again the MVP infection hits Portugal, Spain, and Canada, when on 18 May 2022, with 14, 7, and 13 cases, respectively reported in these countries [6]. The MVP continues to spread to Belgium, Sweden, and Italy when they confirm their first MPV cases. Similarly, on 20 May 2022, Australia reported two cases. One was from Sydney and the other was in Melbourne. With each passing day, the MVP continues to grow rapidly. It’s when Switzerland and Israel confirmed their first cases on 21 May. Belgium introduces a 21-day mandatory quarantine for MVP. Which reflects the seriousness of this possible pandemic [7]. Thus far, the MVP hits more than 50 countries including Denmark, Canada, North America, United Arab Emirates, the Czech Republic, Slovenia, and the Canary Islands.

A cumulative total of 21,099 confirmed cases have been reported as of 28 July 2022 worldwide. Similarly, a single death from MVP has also been reported to WHO from 42 countries in five WHO Regions [8]. The majority of the confirmed cases, i.e., 98% have been reported since May 2022. Adding to the health concerns, the MVP has greatly affected people’s lives as well as the world’s economy. Among such questions, the people’s and government’s main concerns lie in the control of the disease and searching for effective community or country-wide interventions. For this purpose, a valid analysis and modeling of the data on daily confirmed cases and mortalities are required.

Several Mathematical and statistical models and methods are available which have been widely used for observing the behavior of epidemiological diseases and pandemics. Statistical models such as grey forecasting methods [9,10], mechanistic models and methods [11], Neural Networks (NN) [12,13], multivariate linear regression [14], computer-generated simulation models [15], time series models [16], and the Interrupted Time Series (ITS) regression models [17,18] were successfully applied to predict the intensity and behavior of the epidemic disease in near future. Among such models, time series analysis and neural networks are key and more realistic methods to predict the behavior, nature, and future of epidemics. There has been quite extensive literature reporting time series analysis for estimating several future scenarios of different diseases and epidemics. However, epidemics are mainly random phenomena due to which the general spread of the outbreaks is characterized by randomness. Statistical methods cannot be generalized for the prevalence of the epidemic in the future that can capture the randomness of the epidemic. To encounter such a problem, a valid and more acceptable method, the Automatic-Regressive Integrated Moving-Average (ARIMA), has been successfully adopted by practitioners in Health science and other fields for estimating epidemics. In many previous studies, the ARIMA model was used for predicting and assessing the incidence and prevalence of diseases. For example, the ARIMA model was applied for estimating Dengue Fever [19], Malaria [20], Hepatitis [21], Tuberculosis [22], Influenza [23], etc. Further, the same ARIMA model was used for predicting the intensity of the past SARS outbreak. The ARIMA model is widely used for forecasting and prediction because it can account for changing trends, cyclicity, periodicity, and random disturbance in time series.

In the present study, we predicted the cumulative cases of MVP at the top throughout the world via ARIMA and Neural Networks. The appropriate ARIMA models for cumulative cases were identified, and then the number of confirmed cases was predicted for the 10 days The main objective of the present paper is to compare and find the most appropriate predictive model and to provide a realistic estimate for the peak time, the intensity of the epidemic, and a realistic picture of the future behavior of the outbreak. The study provides a road map for the concerned authorities to supply and plan resources effectively to control the epidemic.

## 2. Materials and Methods

### 2.1. Study Area and Data Description

The data for the outcome variable (cumulative confirmed cases) of MVP were taken from the official website of “Our World in Data” [24]. The data of total confirmed cases were obtained from 6 May 2022 to 28 July 2022. The descriptive statistics of the MVP datasets are given in Table 1. For practical and rational modeling through ARIMA, at least 30 observations were required [25]. Therefore, approximately 60 observations from each country were considered to predict the MVP prevalence in the selected countries. The distribution of the MVP cases (having more than 50 cases) were shown in Table 1 [26]. The total cases were forecasted for a period of 10 days, with a 95% relative confidence interval.

### 2.2. ARIMA Models

Time series analysis consists of methods for analyzing and extracting meaningful statistics and other characteristics from time series data [23,27,28,29]. In time series analysis, ARIMA modeling is considered one of the most suitable and promising forecasting techniques for predicting the future. The ARIMA model was first introduced in the 1970s by two statisticians, George Edward Pelham Box and Gwilym Meirion Jenkins [25,30]. Having the ability to assess the different components of the time series such as trends, cyclicity, periodicity, and random disturbance, the ARIMA models are broadly used for time series analysis.

The ARIMA model is generally expressed as ARIMA (p,d,q), where, p is the order of auto-regression, d signifies the difference trend, while q denotes the order of moving average [25].

The Auto-Regressive AR (p) model specifies that the output variable of the time series $Y_{t}$ depends linearly on its previous values $Y_{t-1}+Y_{t-2},\ldots ,Y_{t-p}$ and on the current residuals $\varepsilon_{t}$ (stochastic term), while the Moving-Average MA (*q*) model emphasizes that the output variable $Y_{t}$ linearly depends on the current and its previous residual series (stochastic terms) $\varepsilon_{t-1}-\varepsilon_{t-2},\ldots ,\;\varepsilon_{t-q}$. The AR (p) and MA (q) models can be expressed in Equation (1) and Equation (2), respectively.
(1)$$Y_{t}={\;\varphi }_{1}Y_{t-1}+{\;\varphi }_{2}Y_{t-2}+\ldots +{\;\varphi }_{p}Y_{t-p}+\varepsilon_{t},$$
(2)$$Y_{t}={\;\theta }_{1}\varepsilon_{t-1}-{\;\theta }_{2}\varepsilon_{t-2}-\ldots -{\;\theta }_{q}\varepsilon_{t-q}+\varepsilon_{t},$$
where $Y_{t}$ denotes the observed value of the time series, φ and θ are the parameters of AR and MA models, respectively, and $\varepsilon_{t}$ denotes the value of random shock at time t. Furthermore, the residual terms (stochastic terms) $\varepsilon_{t}$ are assumed to be identically and independently distributed with zero mean and constant variance $\sigma^{2}\;i.e.\;\varepsilon_{t}\sim iid\;\left(0,\sigma^{2}\right)$.

Combing the MA and AR model, a more general form of the Autoregressive-Moving-Average (ARMA) model is developed. Being composed of AR and MA models, the ARMA (p,q) models specify that the output variable of the time series $Y_{t}$ depends linearly on its previous values $Y_{t-1}+Y_{t-2},\ldots ,Y_{t-p}$, as well as on the current residual series $\varepsilon_{t}$ and the previous residual series $\varepsilon_{t-1}-\varepsilon_{t-2},\ldots ,\;\varepsilon_{t-q}$. The ARMA model can be generally represented by the following equation.
(3)$$Y_{t}{\;=\alpha +\varphi }_{1}Y_{t-1}{\;+\;\varphi }_{2}Y_{t-2}{+\ldots +\varphi }_{p}Y_{t-p}{\;+\;\varepsilon }_{t}{-\;\theta }_{1}\varepsilon_{t-1}{-\theta }_{2}\varepsilon_{t-2}{-\ldots -\;\theta }_{q}\varepsilon_{t-q},$$
where α is a constant, and $\varepsilon_{t-1}$ is the previous random shock value. The ARMA model is modified to the ARIMA model to deal with non-stationary time series. The non-stationary time series can be differenced and modeled as an ARMA model to perform the ARIMA model [23].

### 2.3. Methodology of ARIMA Models

The ARIMA modeling methodology consists of four basic iterative steps:

(1) Identification and assessment of the model, (2) parameters estimation of the identified model, (3) diagnostic checking for the appropriateness of the identified model, and (4) prediction for the future, i.e., forecasting. These iterative steps are shown in Figure 1.

In forecasting via ARIMA models, the Auto-correlation Function (ACF) and Partial Auto-correlation Function (PACF) are the most important analytical tools as they measure the statistical relationship between the observations in univariate data series. The autocorrelation function (ACF), as the word auto-correlation makes clear, only finds out the correlation with itself, i.e., with its lag values in the considered univariate time series. More specifically, the ACF describes how well the present value $Y_{t}$ is related to its past values (lag values) $Y_{t-1}+Y_{t-2},\ldots ,Y_{t-p}$, within the same series. While finding a correlation between the values, the ACF considers all four components (trend, seasonality, cyclic, and residuals), which is why the ACF is known as a “complete auto-correlation plot” [31].

The Partial Autocorrelation Function (PACF), unlike ACF, finds the correlation of the residual (retained after the removal of the effects which are already explained by the earlier lag(s) with the next lag value). In PACF, we first remove the variations found in the series and then find the next correlation which is why it is called a “partial” not “complete” auto-correlation plot.

Basically, in PACF, if any hidden information in the residual is left in the series it is modeled by the next lag, hence PACF might obtain a good correlation between the residual with its next lag value. It is noteworthy that, in time series modeling, we avoid too many features which are correlated (may cause multicollinearity) and keep only the relevant features. The PACF plot is used to find out lag values with high correlation, seasonality in the series, and some kind of trend in both the mean and variance of the series [31].

For identification of the initial model for forecasting (Step 1 in ARIMA modeling), ACF and PACF are estimated. The ACF and PACF are not only used to guess the primary model but also used to approximate estimates of the parameters [25]. When the tentative model is guessed in the first step, the next step (Step 2) is to estimate the parameters of the guessed model via Maximum Likelihood Estimation (MLE). Maximizing the probability of the observation, the MLE finds the parameters of the primary model. In the third step (Step 3), the model adequacy is checked through different diagnostic tests. The residuals are assumed to be a white noise process (the residuals themselves are independent and identically distributed (i.i.d) and the process is stationary and independent). Serval diagnostic tests such as L-Jung-Box, Q-test, residual analysis, and histogram of the residuals are performed for checking the assumptions [32]. In this study, we carry out residual analysis through ACF and PACF of the residuals for validating the assumptions.

Once the assumptions are validated then we move to the fourth step (step 4) which is forecasting. However, if these assumptions are violated, the model automatically goes to the first iteration (step 1). Moreover, if there is more than one successful ARIMA model, the best model among them is selected using certain criteria discussed in the next section (Section 2.4 Model selection).

### 2.4. Multilayer Perceptron Network (MLP)

A supervised machine learning model multilayer perceptron model (MLP) which is also known as the Backpropagation network (BPN) is based on the feed-forward neural network algorithm with different activation functions. This model is acknowledged as one of the most dominant and significant models in time series forecasting due to the algorithms used in processing the information. The structure of the model is consisting of the input layer and single hidden layer with k hidden neurons and an output layer. For information processing, this network utilizes two operations, feedforward, and backpropagation. In the feed-forward operation, the inputs are provided in the form of data and this information is passed to the hidden layer whereby using the suitable activation function which results in an output of the network. This information processing network is based on the connecting layers that are disjoint in the network. Mathematically, the network of the multilayer perceptron model is given by the equation
(4)$$W=f_{s}\left(\overset{K}{\underset{k=0}{\sum }}Y_{1k}^{0}\left(f\overset{N}{\underset{n=0}{\sum }}Y_{kn}^{i}u_{n}+B_{n}\right)\right)$$
where the network inputs $u_{n}$, $B_{n}$ is the bias of the network while *f* is the activation function of the intermediate layers, and $f_{s}$ is the output layer activation function. Y is the output signal, W^*i*^_*kn*_ is the weights of the intermediate layer, and Y^0^_1*k*_ is the connections of the output neurons. In the MLP network, the model training is assumed as the process of adjusting the suitable weight to obtain the optimum output, and to perform this task, the backpropagation method is used in most situations.

### 2.5. Model Selection and Accuracy Measures

Several criteria to test the accuracy of the model are available which compare the observed and predicted values. Akaike information criterion (AIC), Bayesian information criterion (BIC), Schwarz information criterion (SIC), Mean Absolute Error (MAE), Mean Absolute Percentage Error (MAPE), Mean Absolute Deviation (MAD), and Root Mean Square Error (RMSE) [33] are widely used. Among these criteria, MSE, RMSE, MAE, and MAPE are selected in the present study, which is shown in Equations (5)–(8).
(5)$$\mathrm{MSE}=\frac{1}{n}\overset{n}{\underset{t=1}{\sum }}e_{t}^{2}$$
(6)$$\mathrm{RMSE}=\sqrt{\frac{1}{n}\overset{n}{\underset{t=1}{\sum }}e_{t}^{2}}$$
(7)$$\mathrm{MAE}=\frac{1}{n}\overset{n}{\underset{t=1}{\sum }}e_{t}$$
(8)$$\mathrm{MAPE}=\frac{1}{n}\overset{n}{\underset{t=1}{\sum }}\frac{\left|e_{t}\right|}{\left|Y_{t}\right|}*100$$
where $Y_{t}$ denotes the observed value at time point *t* of the series, $e_{t}$ is the difference between the observed and estimated values at time point *t*, while n is the number of time points. The minimum is the value of MSE, RMSE, and MAE, MAPE the better will be the fit of the data. All statistical analyses were performed using MS−Excel−360 and “forecast, tseries, and zoo” libraries built in R−4.0.0 software with a statistically significant level of p<0.05.

## 3. Results and Discussion

The daily cumulative samples of monkeypox are collected for analysis purposes. Recommendations on the minimum necessary number of time points for time series analysis vary, however, there is considerable consensus that this minimum requirement is in the middle two-digit range, for instance, “… 40 observations is often mentioned as the minimum number of observations for a time series analysis” [27], “Most time series experts suggest that the use of time series analysis requires at least 50 observations in the time series.” [30]. There are a total of 84 samples that are part of the analysis therefore formal time series analysis can be performed for future forecasting. The analysis begins by making a graph of the monkeypox cumulative cases. The graph of the monkeypox series is presented in Figure 2.

For processing the analysis ahead, we first describe the summary of the monkeypox data the results are shown in Table 2, and then we apply the ARIMA methodology and then we apply the machine learning model. For the ARIMA model, we begin with the first step of the methodology which is the identification of the model, and to achieve this end we begin with the stationary test. For the stationary confirmation, we apply the Augmented dicky fuller test to the series and after confirming that there is no non-stationarity in the series, we make the correlogram which is the plot of ACF and PACF to identify the model (Table 3). By applying the ADF test it is found that the series is not stationary and to make it stationary we apply a different transformation.

From the graphical perspective, it is found that the series has stationarity in nature and by applying the 1st difference it is removed as mentioned in Table 4. Now to proceed with the analysis we will make the ACF and PACF of this 1st difference series to estimate the significant parameter. The correlogram is given below to move on to the second step of this methodology (Figure 3).

By using the order of the correlogram and using the subjective approach we will estimate the significant parameters of the series. The different combination of the candidate model is given in Table 5. From the output, it is found that among the three different classes of models the model ARIMA (7,1,7) is the best fit for the series as it has low values of the accuracy measure so the model is found significant according to the accuracy criteria, we will check the model and apply the diagnostic checking. To this end, we will make the ACF of the residuals and if there is no lag out from the boundary of 95% confidence interval the candidate model seems to be the best and most significant to model the series. The ACF of the candidate model ARIMA (7,1,7) is given in Figure 4. From the ACF plot, it can be observed that no lag exceeds the confidence limits, so the model seems significant in forecasting the series of Monkeypox. Further the actual versus the fitted values from the model ARIMA (7,1,7) are shown in Figure 5.

Actual cases are the observed number of monkeypox cases and fitted cases are those which have been obtained from the ARIMA model. Now the model ARIMA (7,1,7) is used for forecasting purposes. The values with a 95% confidence interval are given below (Table 6). Table 6 points give the forecasted results from the ARIMA model of monkeypox cases for future predictions with their confidence intervals.

### Multilayer Perceptron Model

In this part, the model is used with the different combinations of the input and hidden neurons with a single hidden layer. The sigmoid activation function is used in the single feed-forward hidden layer. The model is selected according to the criteria of accuracy. A different combination of the models for the monkeypox data is given in Table 7. From Table 7 it is found that the model with the single input layer with 10 hidden neurons has the lowest accuracy measures and also the observed versus the fitted values seem quite well, which is given below in Figure 6, further this model is used for forecasting purposes. Forecast values of the MLP model for the monkeypox data are shown in Table 8.

Here, Actual cases are the observed number of monkeypox cases and fitted cases are those which have been obtained from the MLP model. Table 8 points give the forecasted result from the MLP model of monkeypox cases for future predictions with their confidence intervals.

## 4. Conclusions

In this work, the comparative analysis was made using the classical time series model with the machine learning mode. First, in this work, we applied the ARIMA model and found the significant one to forecast the series. From the results, it was found that the monkeypox series followed the ARIMA (7,1,7) model among the other candidate models, with the root mean square error of 150.78. Comparatively, we applied the multilayer perceptron model with a different number of hidden neurons with a single hidden layer that uses the sigmoid activation function. The output of this model using single input with 10 hidden neurons resulted in significantly accurate measurements, as this model had the root mean square error of 54.40, which is much better than the ARIMA model; furthermore, the actual versus the fitted plot confirmed that the multilayer perceptron model had a better fit for the monkeypox data than the ARIMA model. For future work, the extreme learning machine model (ELM) support vector machine (SVM) and other unorganized methods with different activation functions can be applied for a better fit. In the light of conclusion drawn from the study, it can be stated that this new monkeypox pandemic is alarmingly increasing in different countries where these cases have been reported. An effort was made to select a suitable model, which will help the authorities to adopt the proper measures for minimizing its effects. If the respective management is unable to stop or reduce the transmission, the entire world may be faced with yet another catastrophe on the level of public health. More importantly, this study provided a comparison of two different forecasting methods and observed that the MLP model is the most reliable forecasting model by comparing it with conventional models. However, the main limitation which can be faced is that the comprehensive study of forecasting this pandemic is still challenging due to the lack of complete data from each country. Therefore, efforts should be made to gather the complete dataset images from the whole world in order to detect its future effects using deep learning or artificial intelligence.

## Figures and Tables

**Figure 1 jcm-11-06555-f001:**
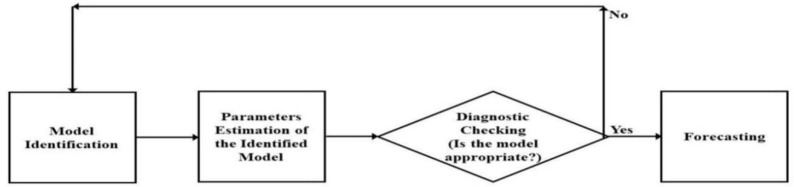
The four iterative steps of ARIMA models for forecasting.

**Figure 2 jcm-11-06555-f002:**
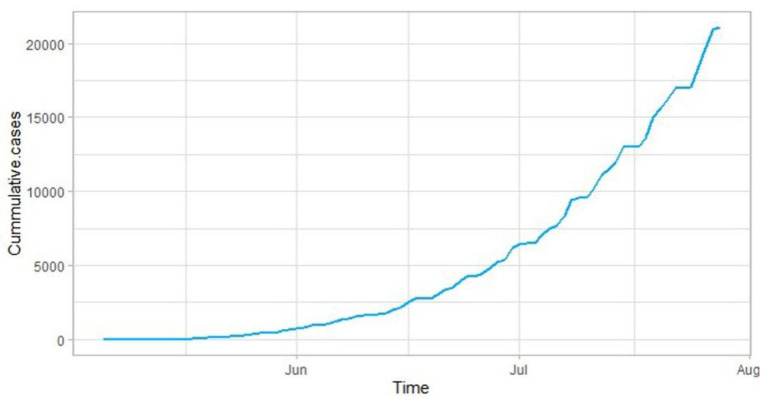
Historigram of the cumulative cases of monkeypox data.

**Figure 3 jcm-11-06555-f003:**
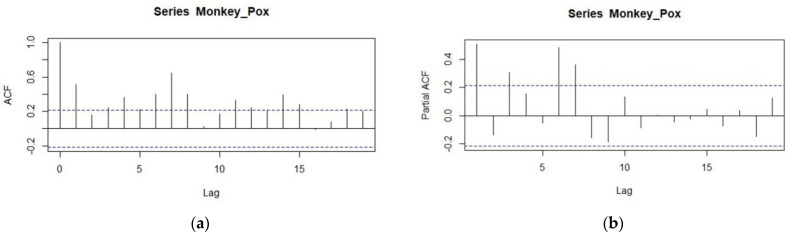
Correlogram of the monkeypox for 1st difference. (**a**) ACF; (**b**) Partial ACF.

**Figure 4 jcm-11-06555-f004:**
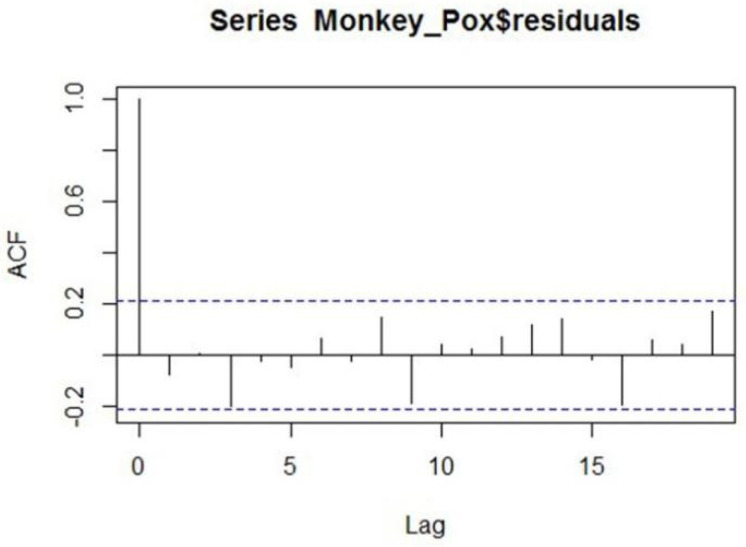
Autocorrelation plot of residuals from ARIMA (7,1,7).

**Figure 5 jcm-11-06555-f005:**
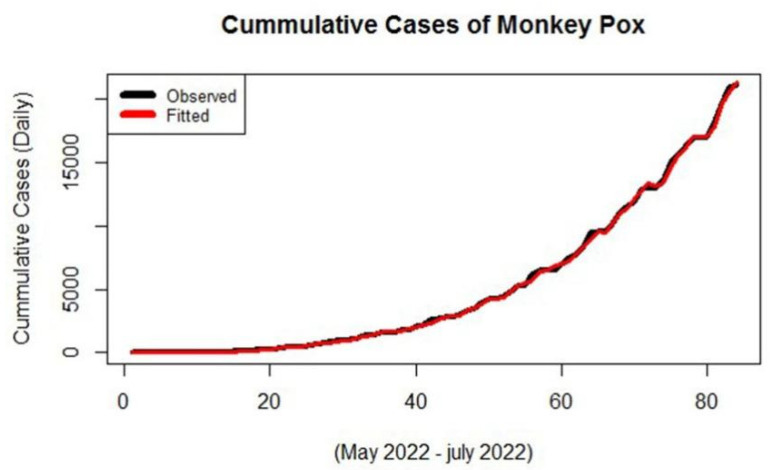
Actual versus fitted plot of ARIMA (7,1,7) for monkeypox data.

**Figure 6 jcm-11-06555-f006:**
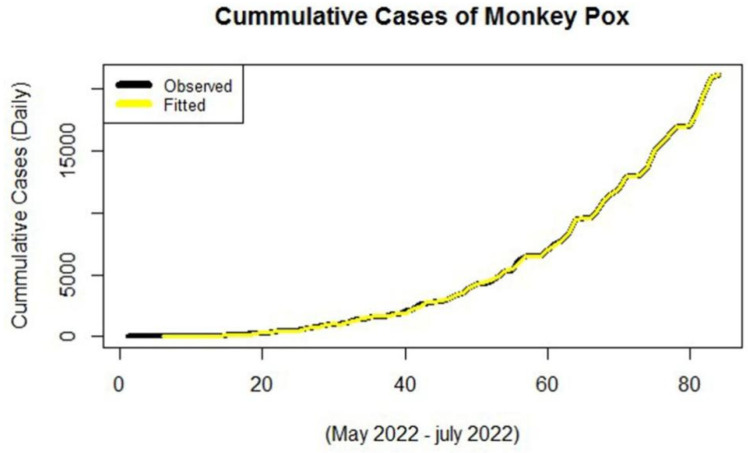
Actual versus fitted plot of MLP for monkeypox data.

**Table 1 jcm-11-06555-t001:** 2022 MPV global outbreak (having more than 50 cases).

Country	Cases	Category
United States	4638	Has not historically reported monkeypox
Spain	3738	Has not historically reported monkeypox
Germany	2459	Has not historically reported monkeypox
United Kingdom	2432	Has not historically reported monkeypox
France	1837	Has not historically reported monkeypox
Netherlands	818	Has not historically reported monkeypox
Canada	745	Has not historically reported monkeypox
Brazil	696	Has not historically reported monkeypox
Portugal	588	Has not historically reported monkeypox
Italy	426	Has not historically reported monkeypox
Belgium	393	Has not historically reported monkeypox
Switzerland	251	Has not historically reported monkeypox
Peru	224	Has not historically reported monkeypox
The Democratic Republic of the Congo	163	Has historically reported monkeypox
Israel	121	Has not historically reported monkeypox
Nigeria	117	Has historically reported monkeypox
Austria	115	Has not historically reported monkeypox
Ireland	85	Has not historically reported monkeypox
Sweden	81	Has not historically reported monkeypox
Denmark	71	Has not historically reported monkeypox
Mexico	59	Has not historically reported monkeypox

Total number of cumulative cases = 21,099.

**Table 2 jcm-11-06555-t002:** Summary statistics for the monkeypox pandemic.

Min	1st Quartile	Median	Mode	3rd Quartile	Max
1	401	2654	5218	8657	21,099

**Table 3 jcm-11-06555-t003:** Augmented Dickey–Fuller test.

Data: Monkey_pox
Dickey−Fuller=3.866, Lag order=4, p value=0.99
alternative hypothesis: stationary

**Table 4 jcm-11-06555-t004:** Augmented Dickey–Fuller test.

Data: Monkey_pox
Dickey−Fuller=−6.8733, Lag order=4, p value=0.01
alternative hypothesis: stationary

**Table 5 jcm-11-06555-t005:** Candidate model for monkeypox using Box–Jenkins methodology.

Candidate Model	MSE	RMSE	MAE	MAPE
ARIMA (5,1,5)	38,549.4	196.34	118.05	6.52
ARIMA (6,1,5)	25,766.67	160.52	94.55	6.29
**ARIMA (7,1,7)**	**22,734.61**	**150.78**	**88.65**	**5.72**

**Table 6 jcm-11-06555-t006:** Forecast values of the model ARIMA (7,1,7) for the monkeypox data.

Serial No	Forecasted Values	Upper 95% C. I	Lower 95% C. I
1	21,516.89	21,845.83	21,187.94
2	21,667.12	22,147.57	21,186.67
3	22,137.39	22,724.06	21,550.72
4	23,283.64	23,977.30	22,589.98
5	24,843.72	25,670.73	24,016.71
6	25,930.43	26,834.66	25,026.20
7	25,916.84	26,834.66	24,902.92
8	26,021.02	26,930.75	24,738.57
9	26,474.18	27,303.47	24,930.92
10	27,300.65	28,017.44	25,559.52

**Table 7 jcm-11-06555-t007:** Candidate model for monkeypox using multilayer perceptron methodology.

Candidate Model	MSE	RMSE	MAE	MAPE
With 5 hidden neurons	6964.31	83.45	56.70	0.27
With 7 hidden neurons	3895.64	62.41	41.66	0.19
**With 10 hidden neurons**	**2960.29**	**54.40**	**32.59**	**0.12**

**Table 8 jcm-11-06555-t008:** Forecast values of MLP model for the monkeypox data.

Serial No	Forecasted Values	Upper 95% C. I	Lower 95% C. I
1	21,124.99	21,960.54	21,222.59
2	21,856.00	22,859.63	22,454.63
3	21,830.08	23,597.83	23,182.77
4	21,926.20	24,765.09	24,295.99
5	21,704.02	24,806.16	25,108.36
6	22,317.85	25,757.07	25,167.07
7	22,507.93	25,046.78	24,846.67
8	22,722.82	26,909.01	26,709.41
9	22,950.31	27,886.04	27,186.14
10	24,269.96	27,995.00	27,885.02

## Data Availability

All relevant data are within the manuscript and its Appendix A.

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
