# Peer review of "Modeling and Forecasting Monkeypox Cases Using Stochastic Models"

_jcm, 2022, doi:10.3390/jcm11216555_

Round 1

Reviewer 1 Report

·    The subset of data may be too small since it only comprised of a few months worth of data that the group is studying. I am not sure if this is acceptable as the references for the modelling  of Dengue, Malaria, Hepatitis and tuberculosis quoted in the submitted manuscript used the surveillance data from a number of years. Hence more appropriate references should be infer using shorter period of surveillance so that it can be comparable to what is proposed in the submitted manuscript.

·   Because the subset is small in the case of monkeypox cases for the submitted manuscript, the cumulative cases for both the actual and predictive seemed to fit. But this fit may not be real if the timeline is prolonged and it  may be neccessary to include data up to the current date. Or compare with some past data if it is available.

·        The researchers who modelled other diseases also used more than one modelling method  and I would recommend the same for the submitted manuscript. This may validate the fittness raised in the previous point.

·        There were no description on the data since ARIMA is supposed to predict on the trends, cyclicity, periodicity; no decription on the advantage that machine learning offers over ARIMA; nor explanation about any limitations in this study. Would suggest that the authors spend more efforts to describe what their predictive models can achieve.

Author Response

Please find our detailed reply in the author letter.

Author Response

(The authors gave the same response as above.)
